# Impact of Initial Population Density of the Dubas Bug, *Ommatissus lybicus* (Hemiptera: Tropiduchidae), on Oviposition Behaviour, Chlorophyll, Biomass and Nutritional Response of Date Palm (*Phoenix dactylifera*)

**DOI:** 10.3390/insects14010012

**Published:** 2022-12-22

**Authors:** Nasser Al-Abri, Suad Al-Raqami, Maryam Al-Hashemi, Rashid Al-Shidi, Salim Al-Khatri, Rumiana V. Ray

**Affiliations:** 1Directorate General of Agriculture and Livestock Research, Ministry of Agriculture, Fisheries Wealth & Water Resources, Muscat P.O. Box 50, Oman; 2Division of Plant and Crop Sciences, School of Biosciences, University of Nottingham, Sutton Bonington Campus, Loughborough LE12 5RD, UK

**Keywords:** *Ommatissus lybicus*, chlorophyll, plant nutrition, date palm biomass, honeydew quantification, oviposition behaviour

## Abstract

**Simple Summary:**

The Dubas bug (*Ommatissus lybicus*) is an economically important insect pest on date palms grown in the Middle East and North Africa. This work aimed to determine at what population density the pest caused the most significant losses in chlorophyll, nutritional composition, or biomass of date palms, and how losses or pest infestation can be predicted using insect honeydew secretion. The oviposition behavior of females at different densities of *O. lybicus* was also investigated. Our results show that populations exceeding 300 nymphs per palm seedling reduced chlorophyll and fresh palm biomass, whilst insect feeding of 300–600 individuals most consistently decreased calcium, magnesium, potassium, and phosphorus of palms. *Ommatissus lybicus* females laid less eggs on palms as insect numbers reached 600 per palm. Honeydew droplets and the magnesium content of palm leaflets allowed the prediction of chlorophyll at different insect densities. Eggs oviposited on the rachis was the best variable to explain reductions in rachis fresh weight biomass. The optimum density of the pest at which losses can occur was estimated at 3–6 nymphs/leaflet and Mg is one of the nutrients that can be used to sustain chlorophyll and increase palm tolerance to pest infestation.

**Abstract:**

The Dubas bug (*Ommatissus lybicus*) is an economically significant pest of date palms. In this study, the effect of the population density of *O. lybicus* on chlorophyll, measured by the soil plant analysis development (SPAD) chlorophyll meter, palm biomass, and the nutritional composition of date palms, were investigated. A further objective was to determine significant relationships between the population density of *O. lybicus*, the number of honeydew droplets, and oviposited eggs. Reductions of up to 8–11% and 29–34% in chlorophyll content and plant biomass, respectively, were caused by infestations exceeding 300 nymphs per palm seedling. Increasing the population density of *O. lybicus* to 600 insects per palm decreased oviposition by females, suggesting intraspecific competition for resources. There was a significant relationship between honeydew droplets produced by the pest population and chlorophyll content in the rachis, suggesting that treatment can be triggered at 3–6 nymphs/leaflet. Egg oviposition was preferentially on the rachis. Ca, Mg, K, and P were the main nutrients affected by the activity of the pest. Mg content was associated with reduced chlorophyll content under increasing pest density, suggesting that supplemental nutrition can be potentially utilized to sustain chlorophyll and increase palm tolerance to pest infestation.

## 1. Introduction

Date palm (*Phoenix dactylifera* L.) is an economically important crop in the arid and semi-arid regions of the Middle East and North Africa [1,2]. Dates are a rich source of sugars, with a sugar content of 72–88% [3], and natural fibre [4], for human nutrition. The area of date palm plantations comprises 60% of the global cultivation in North African and Middle East countries, with nearly 60 million date palms [5].

The Dubas bug (*Ommatissus lybicus* de Bergevin) (Hemiptera: Tropiduchidae) (Figure 1) is a phloem-feeding insect-pest characterised by modified mouthparts containing needle-like stylets, which allow feeding by piercing date palm leaf tissue [6,7,8].

Feeding disrupts photoassimilate flow and hinders phloem transport by interfering with sugar and nitrogen delivery to cells, as in the case of aphids [6,9]. Endophytic oviposition, probing, and salivation during feeding result in anatomical changes and mechanical injury to palm tissues [10,11,12,13]. Severe insect infestations can cause up to 50% date palm yield loss [12,13,14] and are associated with significant reductions in fruit size, sweetness, and delayed date ripening [15,16,17,18]. As a bivoltine, *O. lybicus* produces a spring generation (February to May) and an autumn generation (August to November) per year [2,19]. The insect has five nymphal instars and can live after hatching for up to 155 days in the autumn and 162 days in the spring [20]. The economic threshold of at least one nymph of *O. lybicus* per leaflet, determined by Arbabtafti et al. [21], has been shown to be inconsistent as it was significantly influenced by geographical location, date palm variety, and environmental conditions.

*Ommatissus lybicus* ingests large amounts of sugar-rich palm sap to obtain amino acids required for growth and reproduction, and excretes the surplus sugars [22]. The substantial secretion of honeydew results in the growth of the black sooty mould, *Meliola camellia* [19,23,24,25,26,27], on fronds, contributing to the accumulation of dust on palm fronds and the intercropped plantation below [2]. Honeydew is a water-soluble complex comprising of carbohydrates (mainly sucrose, glucose, fructose, and others), amides, amino acids, alcohol, salts, and auxins [28,29,30,31]. Honeydew droplets secreted by *O. lybicus* can be positively related to insect numbers on the host, and potentially utilised to estimate the size of populations without the need to manually count insects on fronds or leaflets [13].

Insect population growth can be significantly impacted by intraspecific competition [32,33], whereby a population density that cannot be nutritionally supported by the host is regulated by a proportional decrease in reproduction rate and/or an increase in mortality or emigration [33,34]. The effect of the intraspecific competition is reported to affect fecundity in many species [35,36,37,38,39,40]. For example, Luft et al. [41] demonstrated that the average number of oviposited eggs by eugenia psyllid *Trioza eugeniae* Froggatt females on Magenta Lilly Pilly *Syzygium paniculatum* Gaertn leaves was significantly decreased when the insect population increased beyond three insects per leaf [41]. Shah et al. [42] reported that morphological and physiological characteristics of date palm fronds (leaflet thickness, chlorophyll, and frond position) failed to affect the oviposition behaviour of *O. lybicus* females, however, no previous studies have been conducted to address the effect of different population densities of *O. lybicus* on the fecundity of the pest.

Phloem-sap feeding insects reduce plant chlorophyll [43,44,45] and Shah [24] previously associated significant chlorophyll reduction of date palm leaflets with 30 insects/leaflet. Numerous studies have indicated a significant impact of plant phloem-feeders on plant biomass [46,47,48,49], nutritional response [50,51], and photosynthetic rate of their hosts [52]. However, no studies have investigated the feeding effects of *O. lybicus* on the nutritional status of date palms. The objectives of this study were thus to: (i) define the relationships between different populations of *O. lybicus* and the number of oviposited eggs or honeydew droplets on the palm host and (ii) determine the impact of different population densities of *O. lybicus* on chlorophyll, nutritional content, and final biomass of date palms.

## 2. Materials and Methods

Culturing *O. lybicus* and all experiments were performed in the Entomology Laboratory of the Plant Protection Research Centre, Directorate General of Agricultural and Livestock Research, Al-Rumais, the Sultanate of Oman.

### 2.1. Insects and Experimental Procedures

The experiments were conducted in an insect growth room with artificial climate control and pre-set conditions of 25 ± 2 °C with 12:12 h photoperiod [2]. Fluorescent lighting tubes (GE Starcoat T5 f28w/865 (6500 K)) with automated timers were used in the room for photoperiod maintenance. Two-year-old tissue culture-derived date palms, cultivar Khalas, were used as host plants. Khalas was selected due to being the most widely grown variety in Oman [53]. The palms were planted in pots (8 L) containing peat moss as a growing medium and were confined inside well-ventilated insect-proof net cages (W 45.5 cm × L 60 cm × H 90 cm). The experiment was designed as a randomized complete block design and consisted of five treatments: control (no insects), 100, 300, 600, or 1000 *O. lybicus* individuals per palm. Treatments were chosen to achieve low (approx. 1 insect/leaflet), medium (3 insects/leaflet), high (6 insects/leaflet), or extremely high (10 insects/leaflet) population densities based on infestation categorization used previously by Al-Khatri [2], calculated per palm, with an average of 5 fronds and 20 leaflets, as used in our experiments.

Treatments had four replicates, and each caged pot was considered a single experimental unit. Immature *O. lybicus* (1st and 2nd instars) were collected from an infested date palm orchard and used for the experiments (23°21′20.9″ N 57°38′09.0″ E). The palms received 20:20:20 + TE NPK fertilizer two weeks before releasing the insects, and were irrigated evenly twice a week. All experiments were repeated twice.

### 2.2. Chlorophyll Measurement

The chlorophyll of palm seedlings was measured using a SPAD-502 plus chlorophyll meter (Konica Minolta Sensing, Inc., Osaka, Japan) [54,55,56,57,58]. Three SPAD readings were taken on ten leaflets of each frond per palm for eleven weeks following insect release. The mean SPAD reading per experimental unit (palm) was used for chlorophyll analysis.

### 2.3. Honeydew Droplets Count

Yellow, water-sensitive papers (26 mm × 76 mm) were used to mark the honeydew droplets secreted by *O. lybicus.* Four papers were placed in four different directions under each palm for 2 h (from 8 a.m. to 10 a.m.) [18]. Honeydew droplet visual counts were carried out once a week, over eleven weeks following insect release in the cages.

### 2.4. Ommatissus lybicus Oviposition Density

After obtaining the last set of chlorophyll and honeydew data, *O. lybicus* egg numbers oviposited on all leaflets and rachises of each date palm were counted to assess the oviposition density in different treatments. Palm tissues were examined with a stereoscope ZEISS SteREO Discovery. V20, (Carl Zeiss Microscopy GmbH, Jena, Germany). Oviposition density was determined by dividing the number of oviposited eggs by the palm surface area.

### 2.5. Plant Area and Biomass Measurements

After procuring the last data set at eleven weeks following insect release, date palms were removed from the growing containers and washed with water to remove the peat moss from the roots and left to dry for one hour. Date palms were then separated into three parts (leaflets, roots, and rachis) to measure their fresh weight. Leaflets were scanned using a Canoscan LIDE120 Flatbed Scanner (CANON), and a known reference scale (10 cm ruler) was added for further image analysis. Image J (https://imagej.nih.gov/ij/, accessed on 17 June 2021) was used to process the scanned images and obtain the leaflets area. Rachis surface area was calculated as a cylinder, by measuring the length of the rachis and the average radius of both ends. Measured individual rachises were then summed to obtain surface area estimates [59,60,61].
A=2πrhwhere A, r, h, and π denote area, radius, height, and Pi (3.141592), respectively.

Each biomass partition was placed into pre-heated (70 °C for one hour to avoid moisture intrusion) paper bags and weighed individually using a digital sensitive balance Kern EG 220-3NM (Kern and Sohn, Germany). The palm tissues were dried using an air oven (Binder GmbH, Tuttlingen, Germany) set at 70 °C for three days, and were immediately weighed again. The biomass data is presented as dry weight and fresh weight.

### 2.6. Plant Nutritional Measurements

Palm tissues were used to investigate nutritional variations between infested and non-infested plants. Initial screening of samples of date palms infested with 1000 *O. lybicus* individuals and the control palms (no infestation) was first performed to determine if any nutrients were affected by extremely high infestation with *O. lybicus* [62]. This was achieved by digesting 0.5 g of the dried material in a 100 mL digestion tube containing 3–4 anti-bumping granules, 3 g catalyst mixtures, and 10 mL concentrated H_2_SO_4_, which was mixed well using a vortex tube stirrer. The mixture was first heated at 100 °C for 20 min in a block-digester, agitated thoroughly, and then heated at 380 °C for two hours. Following digestion, the mixture was cooled down, made to 100 mL volume by adding distilled water, and filtered using Whatman filter no. 42 (Whatman Ltd., Maidstone, UK). Nitrogen was analysed using a Kjeltec 2400 distillation unit (Foss Analytical, Hillerød, Denmark). Ammonium-vanadomolybdate (10 mL) was added to 10 mL of the fluid, in a 100 mL flask [62]. K and Na were analysed using the dry ash method. Ground plant materials were weighed (1 g) into 50 mL porcelain crucibles and heated in the furnace to 550 °C for 5 h. After heating, the crucible was allowed to cool, and 5 mL portions of 2 N HCl were added to the ash before mixing thoroughly. After 20 min, distilled water was added to make 50 mL, before filtering the mixture after 30 min using Whatman filter no. 42, and performing analysis using a flame photometer (Jenway PFP7, Stone UK). The remaining elements (B, Ca, Cu, Fe, Mn, Mo, S, Zn, and Mg) were analysed using inductively coupled plasma (ICP) spectroscopy (ICPE-9000, Shimadzu, Japan) [62]. Nutrient analysis, as described above, was subsequently performed on all samples from the first and the second experiments to confirm the main nutrients affected by different densities of insect infestation.

### 2.7. Statistical Analysis

Statistical analysis was performed using Genstat 22nd edition. Chlorophyll and honeydew droplets were analysed using a generalized linear mixed model [63]. Normal distribution with no transformation was used to model chlorophyll with function identity, as residuals for chlorophyll data were visually confirmed to show normal distribution. Negative binomial distribution with link function of log ratio was used to model honeydew droplets. Nested palm, frond, and leaflet were included as random effects and experiment, time (weeks following infestation), and treatment (insect density), and all interactions were included as fixed effects in both models. Final models excluded fixed terms which were not significant, and included the interaction (time × treatment) as the fixed effect, with time as a continuous variable. Oviposited eggs were analysed using a generalized linear model (GLM) with block, experiment, treatment (insect density), and experiment*treatment interaction as fixed effects. Binomial distribution with link function of logit was used to model oviposition site and oviposition on leaflets and rachis (as proportion of total eggs per palm area in cm). All estimates were back-transformed, showing standard errors of the mean. Fisher’s least significant difference at *p* < 0.05 was used for multiple comparisons following GLM for egg oviposition on leaflets and the rachis. Plant nutritional values for the initial screen comparing the control (no infestation) and infestation of 1000 insects were analysed using *t*-test at 95% confidence level. Plant nutritional values and biomass for the full experimental treatments were analysed using analysis of variance (ANOVA), and the least significant difference at *p* < 0.05 was used to compare means between treatments. Single or multiple linear regressions with groups for treatment (insect density) were used to predict chlorophyll and rachis fresh weight. Regression models tested if data fitted common, parallel, or separate lines for treatment groups before final model selection at *p* < 0.05.

## 3. Results

### 3.1. Loss of Chlorophyll

General linear mixed model analysis revealed significant interaction (*p* < 0.001) between time of measurement (weeks) and treatment (population density) for chlorophyll content (Figure 2).

A significant chlorophyll loss occurred in response to an increasing population beyond 300 *O. lybicus* per palm from week seven onwards. At the end of measurements, the population of 600 insects caused the highest SPAD loss of 11%, followed by 8% for densities of 1000 and 300 insects, respectively (Figure 2).

### 3.2. Honeydew Droplets

A significant interaction between time and population density was observed for honeydew droplet secretion (*p* = 0.049, Figure 3).

The highest honeydew droplets of 78.9 and 90.8 were secreted by populations of 1000 and 600 insects, respectively. The population of 300 insects produced half of the honeydew droplets compared to populations of 600 or 1000 insects. The honeydew droplets decreased significantly following the third week of insect infestation (Figure 3).

### 3.3. Ommatissus lybicus Oviposition

There was a significant difference for oviposition site (rachis or leaflets) with insects preferentially ovipositing on the rachis rather than on palm leaflets with a ratio of 2:1 (*p* < 0.001, Figure 4).

There were significant interactions between experiments and population densities, with *O. lybicus* at lower densities of 100 and 300 ovipositing significantly more eggs than at higher densities in experiment 1 (Figure 5). In contrast, in experiment 2, oviposition increased with insect density reaching 80% of total egg oviposition on the rachis by 600 insects per palm.

A significant interaction between experiment and population density was also observed for egg oviposition on leaflets (*p* < 0.001, Figure 6). The maximum egg oviposition on leaflets was by 600 insects in experiment 1 and 100 *O. lybicus* in experiment 2 (Figure 6).

### 3.4. Plant Biomass

There were no significant interactions between experiment and treatment for plant biomass of the rachis, leaflets and roots; however, fresh and dry biomass of roots was significantly higher in the second experiment compared to the first (Table 1). Fresh weight of leaflets was significantly reduced by 29% and 22% by insect infestations of 300 or 600 *O. lybicus*, respectively (Table 1). Infestations beyond 300 insects also caused up to 34% reduction in the fresh biomass of the rachis compared with the control. Fresh weight of roots was reduced by insect populations exceeding 100 individuals, with 600 insects per palm reducing root weight by 34%. There were no significant differences detected for the dry weight of leaflets or rachis. The highest root dry weight reduction of 32% was caused by the population density of 300 insects in the first experiment; however, in the second experiment, the population density of 600 insects reduced root dry weight by 28% compared to the control (no insects).

### 3.5. Nutritional Measurements

The nutritional screen with 13 elements including N, P, K, B, Ca, Cu, Fe, Mn, Mo, Na, S, Zn, and Mg on material from palms exposed to 1000 insects or the control revealed that *O. lybicus* only affected Mg, P and K content in leaflets, rachises, or roots (Table 2).

Nutrient analysis of all samples in experiments 1 and 2 confirmed that the concentration of the same elements as shown in the screen, in addition to Ca, was affected by insect infestation. Significant interactions between the experiment and treatment were observed for K and Mg in the leaflets, and P, in both the leaflets and the rachis (Table 3).

Ca content in leaflets was lower in the first experiment compared to experiment two with any infestation causing a significant reduction compared to the control. The lowest Ca concentration of 4.2 mg/kg was quantified in palms with the starting population of 600 insects. K concentration in leaflets was lower in experiment two and increased with infestation up to 600 individuals, but in experiment one, 1000 *O. lybicus* reduced K compared to the control. Mg in leaflets was lower in infestations of 1000 insects in experiment one, and was reduced by 100 and 600 insects in experiment two. The effect on P in leaflets by insect density was also inconsistent between experiments, being higher in experiment one, with a reduction only seen with an infestation of 1000 nymphs. In contrast, in experiment two, the greatest reductions of P were observed at the lower infestations of 100 to 300 insects. P in the rachis and K in the roots were lower in experiment two and were reduced by the infestation of 1000 individuals. Mg in roots was not affected by insect density but was significantly lower in the first experiment.

### 3.6. Regression Analyses

Simple linear regression analysis with the insect population as groups, revealed that data for each population density fitted parallel lines for honeydew droplets in Week 1, and Mg content in leaflets, explaining 48% and 62%, respectively, of the variance associated with chlorophyll content in week 11 (*p* < 0.001, Table 4).

Rachis fresh weight was related to eggs oviposition on the rachis accounting for 38% of the variance, with data fitting common line for *O. lybicus* population density (*p* < 0.001, Table 4).

## 4. Discussion

This study investigated the impact of feeding, honeydew excretion, and egg oviposition by different initial populations of 1st and 2nd instars of *O. lybicus* on chlorophyll, plant nutrients, and biomass of date palm. The results revealed that the infestations by populations greater than 300 nymphs of *O. lybicus* caused significant chlorophyll reduction in leaflets, associated with feeding and sugar drain over the 11 weeks of insect infestation. Several studies have previously reported a negative effect of phloem feeding insects on the chlorophyll of different host plants. Rafi et al. [38] showed that probing by the Russian wheat aphid *Diuraphis noxia* Mordvilko caused a reduction in the total chlorophyll content by 18.2%, at a density of 80 insects/plant on *Triticum aestivum* L. Similarly, Huang et al. [45] reported that an initial population of five nymphs of the invasive mealybug *Phenacoccus solenopsis* Tinsley per plant caused a 57% reduction in chlorophyll content after 38 days of feeding on leaves of *Solanum lycopersicum* L. Previously, Shah [24] exposed three date palm cultivars (Kehraba, Jan Sore, and Mozavati) to different initial *O. lybicus* population densities of 5, 10, 15, 20, 25, and 30 first instar nymphs per leaflet confined to micro cages of 1.9 cm diameter. His results revealed that the chlorophyll of palm leaflets was reduced significantly by 30 insects/leaflet in comparison with the control treatment, and there were no differences at lower insect populations. In contrast to using individual leaflets by Shah [24], we studied the effects of insect populations confined to whole young palms. The effects of *O. lybicus* feeding on date palm seedlings were not immediate, and chlorophyll loss increased significantly beyond the first seven weeks of sustained feeding, causing a maximum loss of 11% with an optimum infestation of 600 insects, roughly equating to a starting population of 6 insects per leaflet. This is in agreement with other authors [43,44,64], showing that adequate exposure time at an optimum infestation population was needed to cause significant chlorophyll content reductions by phloem-feeding insect pests. The time taken for the metamorphosis of nymphs to adults is approximately five weeks for *O. lybicus*, and it is clear from our results that chlorophyll loss of leaflets increased once nymphs morphed into adults, on the palms. Increased loss in chlorophyll over time was associated with reductions in honeydew secretion by populations greater than 300 insects, suggesting nutrient resource exhaustion caused by feeding of the nymphs and diminished ability of the host to recover. Our studies also indicated that date palms have the potential to tolerate the lower infestations of 100 nymphs for up to 11 weeks, because at this population density losses remained lowest for the period of the studies, and were not significantly different from the control.

Mechanical probing and salivation by *O. lybicus* are likely contributing to chlorophyll loss through feeding, as shown with other insect-host systems [10]. Indeed, in our studies, honeydew secretion in week one, following insect infestation, was negatively related to chlorophyll content on leaflets at the end of experimentation. However, endophytic oviposition by *O. lybicus* females, with a saw-like robust chitinous tool around its ovipositor, tear date palm tissue, depositing eggs at 0.4 to 0.5 mm deep in tunnels [2,15], causing acute tissue necrosis, which can potentially decrease not only chlorophyll content but also rachis biomass. This is supported by the observed negative relationship between egg oviposition on the rachis and rachis biomass fitting common line for all insect densities. Shah et al. [42] showed a negative relationship between measured chlorophyll and the oviposited eggs in different frond rows in spring (r = −0.85) or autumn (r = −0.77). In the current study, *O. lybicus* females generally exhibited a preference to oviposit their eggs in rachises rather than in leaflets, with a ratio of 2:1. This mechanism avoids rapid exhaustion of host assimilates as photosynthetically active leaflets are not being destructively used by females for oviposition, but rather to support feeding of the developing nymphs of the next generation. However, at lower densities than 1000 individuals, there were inconsistencies between experiments in egg oviposition on individual tissues. For example, at 100 insects per palm, a greater proportion of eggs were oviposited in leaflets in experiment two than in experiment one, with the opposite trend observed in experiments for 600 insects per palm. It is possible that the oviposition behavior of insects at different densities in the two experiments related to differences in palm quality used in individual experimental repeats.

A decline in reproductive output, due to direct intraspecific competition for food resources, is often observed with increasing population density of insects and animals [35,36,37,39,40]. Chlorophyll loss and egg oviposition by *O. lybicus* on the rachis reached its optimum, with 300 to 600 individuals, before generally decreasing with extremely high infestations of 1000, at the end of experimentation. This is likely to be associated with decreased availability and/or quality of nutrition from the host. Indeed, palms with moderate to high infestation of between 300 to 600 individuals sustained the greatest reductions of biomass. Similarly, Rafi et al. [38] reported that a high initial population density of the wheat aphid *D. noxia* caused a reduced reproduction rate in its wheat host, whilst Mamai et al. [37] showed an adverse effect of increased adult density of *Anopheles arabiensis* Patton on egg production.

Previous studies by Mokhtar & AI-Mjeni [13] and Shah et al. [18] have reported associations between the density of *O. lybicus* and the amounts of honeydew droplets. However, our results suggested that honeydew deposits were related to the initial population density of the nymphs during the very first few weeks of infestation, and once the nymphs morphed into adults there were smaller differences in honeydew deposits between populations. This suggests that honeydew deposits can be utilized to predict early nymph populations associated with chlorophyll loss. We note that the observed relationships with honeydew deposits in our studies were of moderate strength (R^2^ = 0.48), suggesting that other factors, such as environmental conditions, for example, not accounted for here, are also likely to be important. A significant reduction in fresh palm biomass of leaflets was observed in our studies when *O. lybicus* density increased between 300–600 individuals per palm. Similarly, rachis fresh biomass was reduced when populations increased beyond 100 individuals. The observed differences in fresh biomass, but not for dry biomass of leaflets and rachises, suggested that insect infestation reduced water content of palm tissues. We also found significant reductions in root fresh and dry biomass in response to infestations by 300–600 individuals. This supports the view that optimum insect feeding and egg oviposition also cause reductions in sugar flow to the roots via the phloem, thus decreasing root biomass and affecting the overall water use efficiency of the palms.

Four nutrients involved in essential plant growth and photosynthesis, Mg, K, Ca, and P, were affected by infestation with *O. lybicus* on date palm leaflets. Calcium is an element used for plant tissue development, promoting cell elongation and strengthening of plant cell walls [65,66], in addition to being involved in signaling during defense against pests and pathogens [65]. In our experiments, Ca was consistently reduced in both experiments by insect infestation, with the greatest reductions caused by 600 insects which also caused a significant decrease in the chlorophyll of the leaflets. Phosphorous and magnesium play essential roles in plant energy transfer and biochemical processes, including chlorophyll synthesis, respiration, and photosynthesis [67,68]. Potassium is also needed for photosynthesis, phloem translocation of assimilates, and metabolism [69,70,71]. However, in our studies, we observed interactions between experiments and treatments for K, Mg, and P in plant tissues, suggesting that the effects of insect populations on nutrient content were inconsistent under our experimental conditions. The reductions of these nutrients in leaflets by specific insect densities in the second experiment appear to be related to the increased oviposition of eggs in leaflets rather than in rachises, specifically in the second experiment, compared to the first. Of all analyzed elements, only Mg concentration was strongly (R^2^ = 0.62) related to chlorophyll content in leaflets of palms with different *O. lybicus* densities at the end of experimentation, suggesting that Mg can significantly contribute to chlorophyll maintenance of date palms. Previous studies have indicated that adequate amounts of Mg contribute to disease resistance and plant yield [72,73]. To maintain optimum growth [74,75], a date palm requires 1.5–3 kg, 0.5 kg, and 2–3 kg of NPK, respectively, in addition to organic manure of 20–44 kg per year [75,76]. The finding that Mg contributes to chlorophyll content under *O. lybicus* infestation suggests that supplementing Mg to conventional fertilizers can be exploited to reduce losses. Further understanding of the effects of targeted applications of Ca, K, and P to supplement nutrition for increased tolerance can be useful in improving pest management in the field.

## 5. Conclusions

The present study reported for the first time the impact of *O. lybicus* on date palm nutrition and biomass of single young date palms. *O. lybicus* infestation by more than 300 nymphs per palm significantly decreased chlorophyll, plant biomass, and palm nutrients. The effect of the feeding process and oviposition was not immediate, and intraspecific competition for space/food resources between *O. lybicus* individuals adversely affected the number of oviposited eggs as the population density of *O. lybicus* reached 600 individuals. This study showed a significant relationship between honeydew droplets produced by different densities of nymphs and chlorophyll content, suggesting that starting nymph populations of 3–6 nymphs/leaflet, equating to medium to high initial infestation, is likely to result in chlorophyll loss of up to 11%, and can be used to identify potential risk and trigger treatment. High to extremely high infestations of 6–10 nymphs/leaflet (600 or 1000 insects per palm seedling as our treatments) resulted in intraspecific competition and a decline in egg oviposition. Egg oviposition was primarily on the rachis, thus reducing rachis biomass, and was also associated with reduced content of P. Egg oviposition on the leaflets was associated with effects on chlorophyll, and the reduction of P, K, and Mg. Magnesium content in leaflets, together with the population density of the nymphs, were significantly related to chlorophyll content, suggesting Mg as a potential supplement for improved palm nutrition under pest attack. Our study was conducted on date palm seedlings in a controlled setting; therefore, validating these results in a commercial date palm plantation is required. Further in field validation of the relationship between the *O. lybicus* infestation and nutritional losses of date palms would provide information to improve crop and pest management via rational application of targeted fertilizers to increase host tolerance to the pest.

## Figures and Tables

**Figure 1 insects-14-00012-f001:**
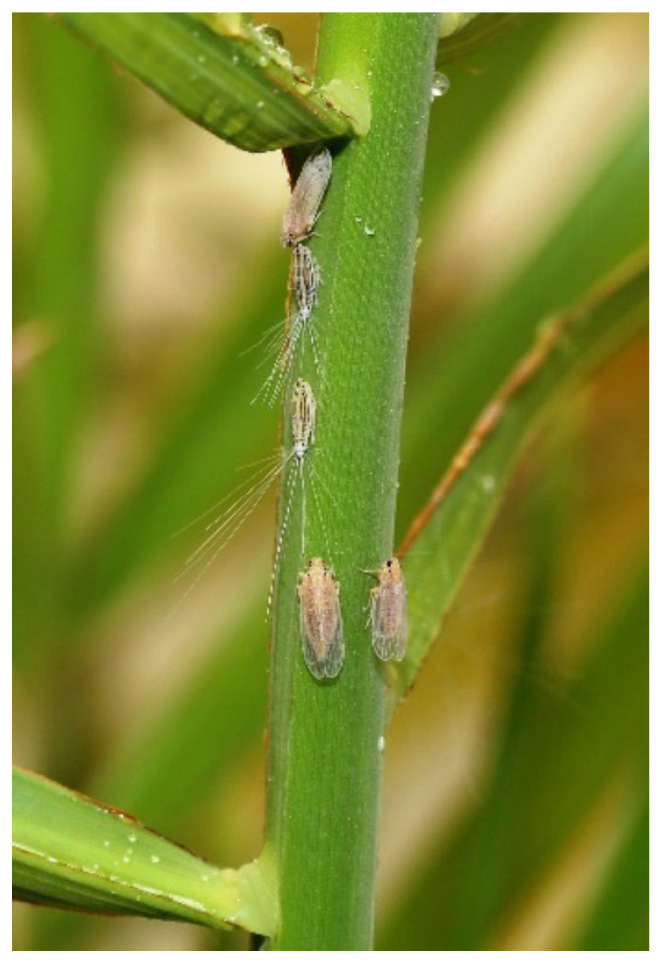
Dubas bug (*Ommatissus lybicus* de Bergevin) nymphs and adults on date palm.

**Figure 2 insects-14-00012-f002:**
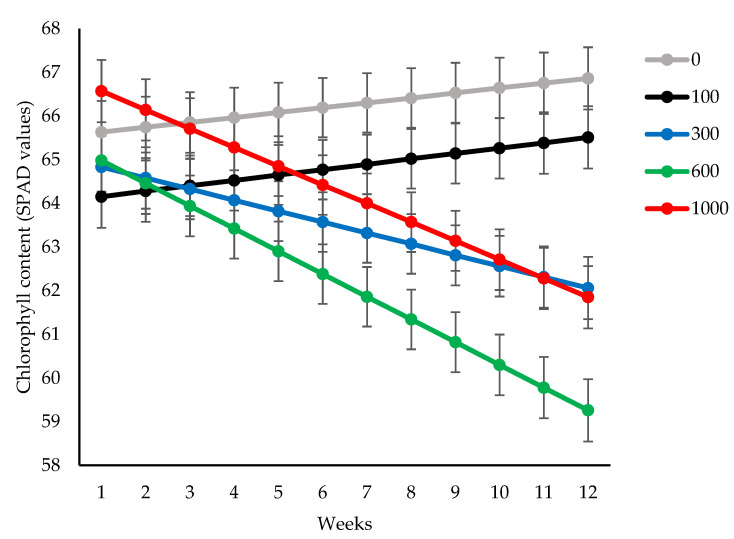
Effect of initial density of *O. lybicus* on chlorophyll measured by SPAD meter on date palm leaflets in the period of eleven weeks following insect release. Predicted means with standard error bars were derived by generalized linear mixed model analysis (*p* < 0.001).

**Figure 3 insects-14-00012-f003:**
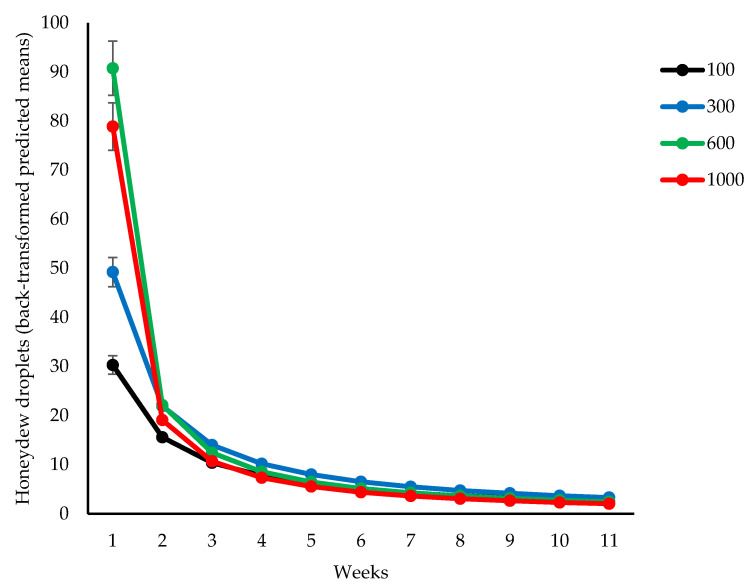
Honeydew secretion by different initial populations of *O. lybicus* in the period of eleven weeks following insect release. Predicted means with standard error bars are shown from generalized linear mixed model analysis for negative binomial distribution with link function of log ratio (*p* = 0.049).

**Figure 4 insects-14-00012-f004:**
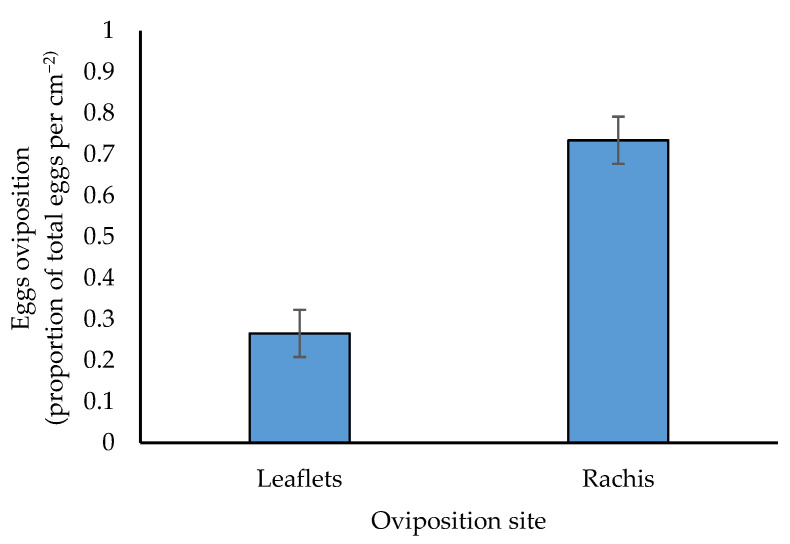
Oviposition site preference (proportion of total eggs per cm^−2^ palm area) of *O. lybicus*. Predicted means with standard error bars were derived by generalized linear model with binomial distribution with link function of logit (*p* < 0.001).

**Figure 5 insects-14-00012-f005:**
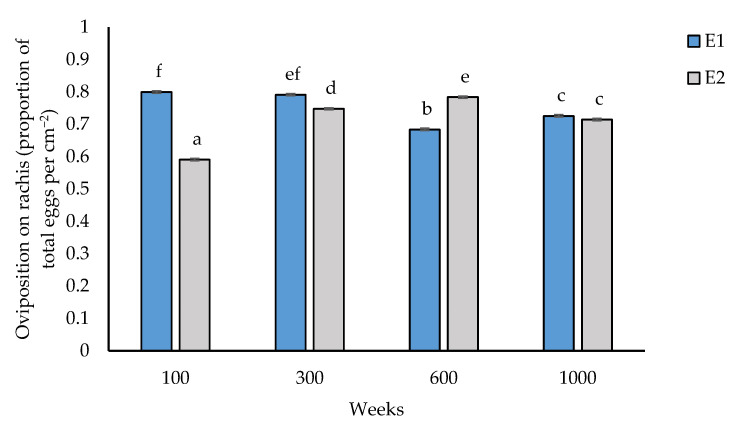
*Ommatissus lybicus* oviposition on rachis (proportion of total eggs per cm^−2^ palm area) in experiments. Predicted means with standard error bars were derived by generalized linear model with binomial distribution with link function of logit (*p* < 0.001). Fisher’s least significant difference test was used for comparisons. Different letters indicate statistically significant differences (*p* < 0.05) between means. Experiment = E.

**Figure 6 insects-14-00012-f006:**
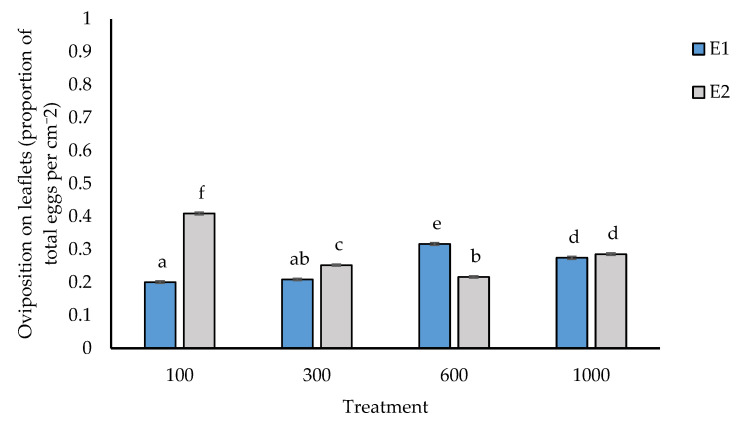
*Ommatissus lybicus* oviposition on leaflets (proportion of total eggs per cm^−2^ palm area) in experiments. Predicted means with standard error bars were derived by generalized linear model with binomial distribution and link function of logit (*p* < 0.001). Fisher’s least significant difference test was used for comparisons. Different letters indicate statistically significant differences (*p* < 0.05) between means. Experiment = E.

**Table 1 insects-14-00012-t001:** Fresh weight and dry weight of the rachis, leaflets, and roots of date palms. ANOVA and the least significant difference at *p* < 0.05 were used to compare means between treatments. T = Treatment, E = Experiment.

N Nymphs	Fresh Weight (g)				Dry Weight (g)			
	Leaflets	Rachis	Roots		Leaflets	Rachis	Roots	
			E1	E2			E1	E2
0	53.4	87.1	51.4	91.8	16.4	16.1	9.6	22.3
100	50.2	73.0	38.1	104.7	15.7	15.8	11.7	28.6
300	37.9	57.2	33.7	69.2	13.7	13.2	6.5	16.8
600	41.6	60.7	36.3	57.6	15.4	14.5	10.8	16.0
1000	48.8	69.1	33.1	66.4	15.7	15.7	9.4	17.5
	*p*-value (LSD)							
T	0.02 (9.9)	0.002 (14.2)	0.008 (17.2)		0.62 (3.5)	0.49 (3.8)	0.03 (5.2)	
E	0.2 (6.2)	0.27 (9.0)	<0.001 (10.9)		0.18 (2.2)	0.25 (2.4)	<0.001 (3.3)	
T × E	0.58 (14.0)	0.55 (20.0)	0.123 (24.3)		0.69 (5.0)	0.55 (5.4)	0.22 (7.4)	
CV%	20.7	19.9	28.8		22.3	24.5	34.2	

**Table 2 insects-14-00012-t002:** Effect of infestation of 1000 *O. lybicus* per palm on nutrient concentration of leaflets, rachises and roots of date palm (cv. Khalas). Only significantly different (*p* < 0.05) elements between treatments are shown.

Number of Nymphs	Nutrient Content (mg/kg of Dry Weight)											
	Leaflets				Rachises				Roots			
	0	1000	*p*-Value	SE±	0	1000	*p*-Value	SE±	0	1000	*p*-Value	SE±
Magnesium (Mg)	3.85	3.48	0.01	0.099	5.65	5.78	0.88	0.759	5.73	4.53	0.02	0.371
Phosphorus (P)	1.38	1.00	0.01	0.095	2.78	2.08	0.02	0.207	1.18	0.98	0.29	0.172
Potassium (K)	9.73	7.53	0.02	0.696	18.33	16.50	0.53	2.734	12.25	9.73	0.01	0.689

**Table 3 insects-14-00012-t003:** Effect of population density of *O. lybicus* (1st and 2nd instar of nymphs) on the concentration of nutrients in leaflets, rachis and roots of date palms (cv. Khalas). ANOVA and the least significant difference at *p* < 0.05 were used to compare means between treatments. T = Treatment, E = Experiment. Only significantly different (*p* < 0.05) elements between populations are shown.

N Nymphs	Nutrient Content (mg/kg of Dry Weight)													
	Leaflets								Rachis		Roots			
	Ca		K		Mg		P		P		K		Mg	
	E1	E2	E1	E2	E1	E2	E1	E2	E1	E2	E1	E2	E1	E2
0	5.5	6.8	9.7	9.1	3.9	4.1	1.4	1.1	2.8	1.9	12.3	11.9	5.7	6.4
100	4.1	5.9	12.2	9.6	4.3	3.5	1.5	0.7	3	1.1	13	11.1	5.7	5.8
300	4.2	6	12.6	9.4	4.3	3.7	1.4	0.8	2.9	1.2	14.1	11.1	5.9	6.2
600	3.6	4.8	12.8	10.4	3.8	3.1	1.3	0.9	2.7	1.2	14.2	11.5	4.8	6.3
1000	4.7	6.2	7.5	10.4	3.5	3.9	1	0.9	2.1	1.5	9.7	11.1	4.5	6.4
	*p*-value (LSD)													
T	0.004 (0.8)		<0.001 (1.0)		0.14 (0.5)		0.2 (0.2)		0.09 (0.4)		0.01 (1.3)		0.5 (0.9)	
E	<0.001 (0.6)		0.03 (1.0)		0.07 (0.3)		<0.001 (0.1)		<0.001 (0.3)		0.04 (1.2)		<0.001 (0.4)	
T × E	0.95 (1.2)		0.008 (1.9)		0.05 (0.6)		0.04 (0.3)		0.05 (0.6)		0.15 (2.2)		0.07 (1.1)	
CV%	17.3		14.7		12		18.5		21.3		15		11.3	

**Table 4 insects-14-00012-t004:** Linear regression analysis of chlorophyll content after eleven weeks of *O. lybicus* infestation and rachis fresh weight with honeydew secretions in the first week, magnesium (Mg) content in leaflets and egg oviposition. (HD1) = (Honeydew 1st week), (ER) = (Eggs oviposited in rachis).

Response Variate (Y)	Equation	R^2^	*p*-Value
Chlorophyll content at 11 weeks	Y (100) = 66.38 − 0.305 (HD1)	0.48	<0.001
	Y (300) = 62.28 − 0.305 (HD1)		
	Y (600) = 59.64 − 0.305 (HD1)		
	Y (1000) = 62.39 − 0.305 (HD1)		
Chlorophyll content at 11 weeks	Y (0) = 1.47 (Mg) + 60.36	0.62	<0.001
	Y (100) = 1.47 (Mg) + 59.49		
	Y (300) = 1.47 (Mg) + 54.78		
	Y (600) = 1.47 (Mg) + 52.28		
	Y (1000) = 1.47 (Mg) + 54.55		
Rachis fresh weight	Y = 87.57 − 5.62 (ER)	0.38	<0.001

## Data Availability

The data presented in this study are available on request from the corresponding author.

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
