# Peer review of "Impact of Initial Population Density of the Dubas Bug, Ommatissus lybicus (Hemiptera: Tropiduchidae), on Oviposition Behaviour, Chlorophyll, Biomass and Nutritional Response of Date Palm (Phoenix dactylifera)"

_insects, 2022, doi:10.3390/insects14010012_

Round 1
Reviewer 1 Report
I was through the Summary, Abstract, Introduction and Materials and Methods. Then I stopped reading the manuscript. The writing style of manuscript is not easy to follow. There are some major flaws in Materials and Methods. Below see my specific comments:
Line 19: Please replace “O. lybicus” with “Ommatissus lybicus”. Remember that scientific names must be in full if they come at the beginning of sentence.
Line 26: Avoid use of abbreviation (e.g., SPAD) in the abstract.
Line 52: What do you mean by “photo assimilate flow”?
Lines 77-78: The sentence needs a rewrite.
Line 91: Replace “the effect of feeding of” with “the feeding effects of”
Lines 92-94: Rewrite the first objective; it is not clear that which variables were compared. e.g., populations of O. lybicus with number of oviposited eggs?, populations of O. lybicus with honeydew droplets? OR number of oviposited eggs with honeydew droplets?
Line 106: Replace “litre” with “liter” due to consistency of using American English.
Lines 108-109: By “randomised block”, do you mean “a randomized complete block design”?
Line 109: Why these densities (100, 300, 600 and 1000 insects) were used? The logarithmic intervals are not equal between densities, as they are multiplied by 3, 2 and 1.67, respectively.
Lines 111-112: “Rewrite the sentence “Immature O. lybicus (1st and 2nd instars) were used and collected from an infested date palm orchard” as “Immature O. lybicus (1st and 2nd instars) were collected from an infested date palm orchard and used for the experiments.
Line 116: You need to mention a general name for “a SPAD-502 plus” (just after the specific name of machine)? For example “a SPAD-502 plus chlorophyll meter”.
Line 129: For how many weeks?
Lines 130-134: It is not clear that what are “sample unit” and “sample size”. How much area on date palms, leaflets and rachises you searched for eggs? How many replications were performed?
Line 135: Corrects the title of section as “Plant leaflet/root/rachis area and biomass”
Line 146: Rewrite the sentence as: Where A, r, h and π denote area, radius, height and Pi (3.141592), respectively.
Line 145: The second part of this equation (2πr2) is measuring sum of top and bottom surfaces of rachis. Why these areas incorporated to the measurement of rachis surface area?
Line 152: Why were the biomass data presented as “a ratio of fresh weight to dry weight”? This is a big surprise to me. We know that both of the fresh weight or dry weight can be used as a measure of biomass. In addition, the dry matter can be calculated as the ratio of dry weight to fresh weight. But I have no idea about the ratio was used here (i.e., a ratio of fresh weight to dry weight). I have looked up into the references given by authors (references 67 and 68). The reference 68 is absolutely irrelevant. I did not find anything close to the ratio used by author (a ratio of fresh weight to dry weight) in the reference 67 as well.
Lines 159-160: I am confused by “were analysed in all treatments in the first and the second experiments”. It is not clear that the 2nd experiment had different treatment from the 1st one. If so, what are the treatment of the 2nd experiment?
Line 187: How many times these treatments were replicated? Or a better question is: It seems that the first experiment is a main set-up and other experiments are just different measurements. Am I right about this? If not, authors need to clearly determine treatments and replications for all of the experiments.
Lines 189-190: Authors stated that “Chlorophyll SPAD values and honeydew droplets were analysed using repeated measurements ANOVA”. Chlorophyll data (as an index of loss) is a proportional data (with binomial errors) but honeydew data are counts (with Poisson errors). So, these two need different data analyses.
Lines 191-193: Again, eggs are count data, plant nutritional values and plant biomass (if the measure used is correct) are proportional data. These must be treated differently for statistical analysis.
Lines 194-196: Needs a rewrite to clear what data were subjected to “stepwise regression” and “multiple regression”. Authors need to give more details about these analyses. In addition, to have a regression the independent variable (populations) must have equal intervals, which is not the case here.
Line 196: The words “the best data fit” is not correct. The correct form is “the best fitted model” or “the best fit”. We know that models are fitted on the data, the reverse is wrong.
Lines 197-199: I did not follow these sentences. They need a rewrite for better understanding.
Author Response
I was through the Summary, Abstract, Introduction and Materials and Methods. Then I stopped reading the manuscript. The writing style of manuscript is not easy to follow. There are some major flaws in Materials and Methods. Below see my specific comments:
The response of the authors to comments of the reviewer is indicated below in red with tracked changes in the reviewed manuscript:
Line 19: Please replace “O. lybicus” with “Ommatissus lybicus”. Remember that scientific names must be in full if they come at the beginning of sentence.
Corrected
Line 26: Avoid use of abbreviation (e.g., SPAD) in the abstract.
Corrected.
Line 52: What do you mean by “photo assimilate flow”?
Photo assimilate flow refers to sugars produced by the process of photosynthesis in the leaves and exported via the phloem. We have clarified in L56-57
Lines 77-78: The sentence needs a rewrite.
L81-84 re-written
Line 91: Replace “the effect of feeding of” with “the feeding effects of”
Corrected
Lines 92-94: Rewrite the first objective; it is not clear that which variables were compared. e.g., populations of O. lybicus with number of oviposited eggs?, populations of O. lybicus with honeydew droplets? OR number of oviposited eggs with honeydew droplets?
Corrected to “define the relationships between different populations of O. lybicus and number of oviposited eggs or honeydew droplets on the palm host
Line 106: Replace “litre” with “liter” due to consistency of using American English.
Corrected
Lines 108-109: By “randomised block”, do you mean “a randomized complete block design”?
Yes. We have added” complete”.
Line 109: Why these densities (100, 300, 600 and 1000 insects) were used? The logarithmic intervals are not equal between densities, as they are multiplied by 3, 2 and 1.67, respectively.
L110-115 We have clarified the rationale behind choosing these densities. Our choice was based on experience working with the pest and previous published work by Al-Khatri (ref included).
Lines 111-112: “Rewrite the sentence “Immature O. lybicus (1st and 2nd instars) were used and collected from an infested date palm orchard” as “Immature O. lybicus (1st and 2nd instars) were collected from an infested date palm orchard and used for the experiments.
Corrected
Line 116: You need to mention a general name for “a SPAD-502 plus” (just after the specific name of machine)? For example “a SPAD-502 plus chlorophyll meter”.
Corrected
Line 129: For how many weeks?
Added 11 weeks following insect release on palms
Lines 130-134: It is not clear that what are “sample unit” and “sample size”. How much area on date palms, leaflets and rachises you searched for eggs? How many replications were performed?
All tissues were examined. Eggs were counted on all leaflets and rachises (on average 20 leaflets and 5 rachises used as sample units), whilst each palm was used as an experimental unit. There were four experimental replicates for each treatment.
Line 135: Corrects the title of section as “Plant leaflet/root/rachis area and biomass”
L140 Corrected to “Plant area and biomass measurements”
Line 146: Rewrite the sentence as: Where A, r, h and π denote area, radius, height and Pi (3.141592), respectively.
Corrected
Line 145: The second part of this equation (2πr2) is measuring sum of top and bottom surfaces of rachis. Why these areas incorporated to the measurement of rachis surface area?
We agree with the reviewer, and we have excluded this part of the equation. We have re-analysed the data based on this and made changes to M&Ms and Results.
Line 152: Why were the biomass data presented as “a ratio of fresh weight to dry weight”? This is a big surprise to me. We know that both of the fresh weight or dry weight can be used as a measure of biomass. In addition, the dry matter can be calculated as the ratio of dry weight to fresh weight. But I have no idea about the ratio was used here (i.e., a ratio of fresh weight to dry weight). I have looked up into the references given by authors (references 67 and 68). The reference 68 is absolutely irrelevant. I did not find anything close to the ratio used by author (a ratio of fresh weight to dry weight) in the reference 67 as well.
L172-173 & L360-382: We have shown fresh weight and dry weight for all experiments, and we have removed the references which are not clearly linked to this work. The ratio of fresh to dry weight reflects the water content of the palms however the difference is also obvious when we use fresh weight data so we can avoid showing the ratio to avoid any confusion.
Lines 159-160: I am confused by “were analysed in all treatments in the first and the second experiments”. It is not clear that the 2nd experiment had different treatment from the 1st one. If so, what are the treatment of the 2nd experiment?
Lines 159-190: Treatments were the same in both experiments. However, an initial screen of nutrient content was carried out on non-infested (control) and material infested with 1000 insects to determine any effect of insect infestation on analyzed elements. Following the screen, which showed that insects affected certain nutrients, all samples from the 1st and 2nd experiments were fully analyzed confirming the results from the screen. Essentially the screen is an additional data set just for the treatment with 1000 insects and control and we have included it in the manuscript. This has been clarified in the text of Materials and methods.
Line 187: How many times these treatments were replicated? Or a better question is: It seems that the first experiment is a main set-up and other experiments are just different measurements. Am I right about this? If not, authors need to clearly determine treatments and replications for all of the experiments.
This is not correct. We have clarified that each treatment had four replicates and the experiment was repeated twice.
Lines 189-190: Authors stated that “Chlorophyll SPAD values and honeydew droplets were analysed using repeated measurements ANOVA”. Chlorophyll data (as an index of loss) is a proportional data (with binomial errors) but honeydew data are counts (with Poisson errors). So, these two need different data analyses.
All data has been re-analysed using general linear regression with poisson distribution for counts. The chlorophyll data set is normally distributed and does not need transformation. Furthermore, we now present the control and have omitted the use of proportion thus SPAD loss index is no longer used in the manuscript. These changes have resulted in major revision of all manuscript sections.
Lines 191-193: Again, eggs are count data, plant nutritional values and plant biomass (if the measure used is correct) are proportional data. These must be treated differently for statistical analysis.
Please see the comment above. Furthermore, we present fresh and dry weight data and have omitted the ratio. Data is normally distributed, and ANOVA was used for the analysis of biomass. All sections have been revised and re-written to reflect the outputs of the analysis.
Lines 194-196: Needs a rewrite to clear what data were subjected to “stepwise regression” and “multiple regression”. Authors need to give more details about these analyses. In addition, to have a regression the independent variable (populations) must have equal intervals, which is not the case here.
For any of the regression analysis in this manuscript population was used as a factor not as an independent variable. For the simple or multiple linear regressions again population density was used as a group thus factor in the analysis. This has been explained in both the materials and methods and the results.
Line 196: The words “the best data fit” is not correct. The correct form is “the best fitted model” or “the best fit”. We know that models are fitted on the data, the reverse is wrong.
Materials and methods, results, and discussion together with conclusions and abstracts were revised and re-written to address all comments of the reviewer.
Lines 197-199: I did not follow these sentences. They need a rewrite for better understanding.
Please see the above comment.
Reviewer 2 Report
The authors have constructed a mostly well-written manuscript addressing an imported pest, Ommatissus lybicus, in date palm production systems in the Middle East and North Africa. These findings will inform O. lybicus integrated pest management programs by reducing unnecessary management activities when infestations are below economically damaging levels.
Comments and suggestions for major revisions are listed below. The quality of tables/charts to enhance presentation/readability could be improved. Justification for data analysis methodology is needed.
Abstract
L31: “…negatively impacted on the oviposition…” Please rephrase this sentence for clarity and grammatical correctness. .
Methods
L189: Please provide more details/clarity on the analytical methods used to analyze the data. What are the fixed and random effects variables? What distribution did you fit the response data to? What were the results and components of the final model? How were the models compared? Currently, the statistical analysis explanations are unacceptable and unable to be reproduced by the reader. This section needs significant revising.
Results
L204: The figures are poor resolution and could be improved. Perhaps use a gray/black color ramp to differentiate lines in “Table 2” and other line chart figures. Also, the authors must label “tables” that are “figures” as such. Include SE bars on bar chart figures.
Discussion
L302: Please speculate on the limitations of the experimental methodology used in this study. For example, only young palms were used, and only controlled experiments were conducted in a laboratory setting. What are the potential future directions of your research to validate these findings in a commercial field setting? Further explain how these results can be used to inform management programs and how these results should be interpreted by growers and the palm industry.
Author Response
The authors have constructed a mostly well-written manuscript addressing an imported pest, Ommatissus lybicus, in date palm production systems in the Middle East and North Africa. These findings will inform O. lybicus integrated pest management programs by reducing unnecessary management activities when infestations are below economically damaging levels.
Comments and suggestions for major revisions are listed below. The quality of tables/charts to enhance presentation/readability could be improved. Justification for data analysis methodology is needed.
The response of the authors to comments of the reviewer is indicated below in red with tracked changes in the reviewed manuscript:
Abstract
L31: “…negatively impacted on the oviposition…” Please rephrase this sentence for clarity and grammatical correctness. .
Abstracts and whole manuscript have gone through major revision to address all comments including English language and grammar checks.
Methods
L189: Please provide more details/clarity on the analytical methods used to analyze the data. What are the fixed and random effects variables? What distribution did you fit the response data to? What were the results and components of the final model? How were the models compared? Currently, the statistical analysis explanations are unacceptable and unable to be reproduced by the reader. This section needs significant revising.
Methods, results, and discussion sections have been re-written to clarify. This is following re-analyzing of the data using linear regression analysis as suggested by the reviewers. Abstracts and conclusions have also been modified to reflect the changes.
Results
L204: The figures are poor resolution and could be improved. Perhaps use a gray/black color ramp to differentiate lines in “Table 2” and other line chart figures. Also, the authors must label “tables” that are “figures” as such. Include SE bars on bar chart figures.
Corrected. We have added SE on bar charts and corrected Figures (erroneously identified previously as Tables). These are numbered consecutively now.
Discussion
L302: Please speculate on the limitations of the experimental methodology used in this study. For example, only young palms were used, and only controlled experiments were conducted in a laboratory setting. What are the potential future directions of your research to validate these findings in a commercial field setting? Further explain how these results can be used to inform management programs and how these results should be interpreted by growers and the palm industry.
L371-453 Addressed as the discussion section has been re-written
L436-458 We provide estimated threshold indicating the potential damage to trigger treatment in addition to nutrient supplements and explain the use of the results for improvement of management of the crop and pest. We further suggest the validation of results in commercial setting.
Reviewer 3 Report
This is an important study on an economically important insect pest of date palms that appears to be well designed and executed. It is well written and results are clearly presented. I have only minor comments:
- It would be probably better to rephrase the title e.g. Impact of initial population density of the Dubas bug, Ommatissus lybicus (Hemiptera: Tropiduchidae), on oviposition behaviour and chlorophyll, biomass and nutritional response of Date Palm (Phoenix dactylifera)
- Lines 18: add 'per palm seedling' after '300 - 600 insects'
- Line 30: add 'seedling' after '300 - 600 insects per palm'
- correct figures captions by exchanging 'Table" to "Figure" and correct numbering of figures and tables
Author Response
The response of the authors to comments of the reviewer is indicated below in red with tracked changes in the reviewed manuscript:
This is an important study on an economically important insect pest of date palms that appears to be well designed and executed. It is well written and results are clearly presented. I have only minor comments:
- It would be probably better to rephrase the title e.g. Impact of initial population density of the Dubas bug, Ommatissus lybicus (Hemiptera: Tropiduchidae), on oviposition behaviour and chlorophyll, biomass and nutritional response of Date Palm (Phoenix dactylifera)
The title has been amended as suggested by the reviewer
- Lines 18: add 'per palm seedling' after '300 - 600 insects'
Addressed: Palm seedling has been used throughout the manuscript
- Line 30: add 'seedling' after '300 - 600 insects per palm'
Addressed. See the comment above.
- correct figures captions by exchanging 'Table" to "Figure" and correct numbering of figures and tables
Corrected
Round 2
Reviewer 1 Report
There is still some main issues with the manuscript as follows:
(1) I am not convinced with the reason the authors gave for my question (Why these densities (100, 300, 600 and 1000 insects) were used?).
(2) Again, I am not convinced with the following answer to my question:
All data has been re-analysed using general linear regression with poisson distribution for counts. The chlorophyll data set is normally distributed and does not need transformation. Furthermore, we now present the control and have omitted the use of proportion thus SPAD loss index is no longer used in the manuscript. These changes have resulted in major revision of all manuscript sections.
What is "general linear regression"? The chlorophyll data are proportional and cannot have a normal distribution.
(3) I am not happy with the following answer as well:
For any of the regression analysis in this manuscript population was used as a factor not as an independent variable. For the simple or multiple linear regressions again population density was used as a group thus factor in the analysis. This has been explained in both the materials and methods and the results.
If population is a factor with different levels, then the corrected analysis is one-way ANOVA or factorial (two-way ANOVA, three-way ANOVA, . . .) depending on the number of factors. So, no regression analysis is needed.
(4) I am not happy with "data analyses" at all.
Reviewer 2 Report
Thank you for providing major revisions throughout the manuscript. The paper is improved.
However, there are still concerns regarding your statistical analyses. Again, this section needs to be clearer and improved.
It would seem that your glm needs to be a mixed model (glmm) to account for the repeated measures experimental design used, correct? Independent time points cannot be modeled as true replicates (only as fixed effects) without accounting for random effects of time using random slopes or random intercept terms, depending on your question/design. You may also need to include random intercept terms for experimental replicates. A glm would be appropriate if you were only testing the significance of O. lyubicus populations individually by week (glm model for each week); overall differences between treatments (across the 11-week sampling period) were not important.
Did you adjust glm analysis for Kenward Roger's degrees of freedom, given the smaller sample size?
Figure 3 is difficult to interpret by placing a factorial variable on the x-axis of a line chart graphic (usually a continuous variable). Please consider visualizing the data another way. Also, the standard error bars are quite small in Figure 3 and oddly consistent; are they correct for individual data points or pooled across treatments?
Author Response
Response by authors in blue below and indicated by ***
Thank you for providing major revisions throughout the manuscript. The paper is improved.
However, there are still concerns regarding your statistical analyses. Again, this section needs to be clearer and improved.
It would seem that your glm needs to be a mixed model (glmm) to account for the repeated measures experimental design used, correct? Independent time points cannot be modeled as true replicates (only as fixed effects) without accounting for random effects of time using random slopes or random intercept terms, depending on your question/design. You may also need to include random intercept terms for experimental replicates. A glm would be appropriate if you were only testing the significance of O. lyubicus populations individually by week (glm model for each week); overall differences between treatments (across the 11-week sampling period) were not important. Did you adjust glm analysis for Kenward Roger's degrees of freedom, given the smaller sample size?
***We thank the reviewer for their comments and suggestions, and we have revised the manuscript accordingly.
L119-214 Chlorophyll and honeydew droplets were analysed using generalized linear mixed model, in Genstat using algorithm by Schall 1991. Normal distribution with no transformation was used to model chlorophyll with function identity as residuals for chlorophyll data were visually confirmed to show normal distribution. Negative binomial distribution with link function of log ratio was used to model honeydew droplets. Nested palm, frond and leaflet were included as random effects and experiment, time (weeks following infestation), treatment (insect density) and all interactions were included as fixed effects in both models. Final models excluded fixed terms which were not significant and included the interaction (time*treatment) as the fixed effect, with time as continuous variable.
Figure 3 is difficult to interpret by placing a factorial variable on the x-axis of a line chart graphic (usually a continuous variable). Please consider visualizing the data another way. Also, the standard error bars are quite small in Figure 3 and oddly consistent; are they correct for individual data points or pooled across treatments?
***Figure 3 has been revised based on the use of glmm as per the previous comment and now shows the back-transformed means from the model with time as continuous variable on the x-axis